

# Quark-gluon tagging: Machine learning vs detector

Gregor Kasieczka[1], Nicholas Kiefer[2], Tilman Plehn[2*], and Jennifer M. Thompson[2]

**1** Institut für Experimentalphysik, Universität Hamburg, Germany
**2** Institut für Theoretische Physik, Universität Heidelberg, Germany

* plehn@uni-heidelberg.de

## Abstract

Distinguishing quarks from gluons based on low-level detector output is one of the most challenging applications of multi-variate and machine learning techniques at the LHC. We first show the performance of our 4-vector-based LoLa tagger without and after considering detector effects. We then discuss two benchmark applications, mono-jet searches with a gluon-rich signal and di-jet resonances with a quark-rich signal. In both cases an immediate benefit compared to the standard event-level analysis exists.

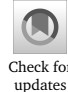
---

**Content**

---

## 1  Introduction

Since the start of the LHC our view of jets as analysis objects has fundamentally changed. While jets with reconstructed 4-momenta matching hard partons still serve as the key objects

of essentially all analyses, their internal structure can now be exploited systematically. In that sense, jets merely define the boundary between event-level observables and subjet observables. The subjet aspect is currently undergoing a paradigm change: rather than defining high-level kinematic observables for the jet constituents and analyzing them using multivariate methods, we can use modern machine learning approaches to analyze low-level detector outputs like the measured 4-vectors entries directly [1]. For this low-level input we employ modern machine learning techniques, usually advertized with the term *deep learning*.

Theoretically and experimentally well-controlled applications of machine learning in subjet physics include hadronic $W/Z$-jets [2–10], Higgs jets [11, 12], top jets [13–20], or model-independent searches for hard new physics in jets [21] quark–gluon discrimination has a long history [22–31] and is used at the LHC [32–34]. However, distinguishing quark and gluon jets poses serious theoretical and simulational challenges, like, that they are not defined in QCD beyond tree level [35–44]. Nevertheless, efficient machine learning approaches have been devised to separate 'quark jets' from 'gluon jets' [45, 46, 48–54]. One way we can overcome the fundamental problems in defining quark and gluon jets is to instead ask for a well-defined hypotheses in terms of LHC signatures, involving mostly gluons vs gluons in the signal and background processes [55–58].

Before we employ modern machine learning to separate processes with mostly hard quarks from those with mostly hard gluons we review the known high-level variables. Unlike for many other subjet analyses these observables rely on tracking information with its excellent resolution, and cannot be considered infrared-safe observables or easily interpretable in perturbative QCD [35–42]. When we switch to low-level inputs this means that we cannot hope for the calorimeter resolution to provide a generous binning and to render us insensitive to additional detector effects. Moreover, any promising network architecture needs to combine standard calorimeter images and tracking information with its vastly better angular resolution [45]. We will use our 4-vector-based LoLa framework developed for top tagging including calorimeter and tracking information [16] to extract the necessary information from measured particle-flow objects and to quantify the sensitivity to soft tracks in the detector. The latter is especially relevant when we benchmark the machine learning approach compared to a multivariate analysis of the traditional quark–gluon variables. In Sec. 2 we analyze idealistic, pure quark and gluon samples to benchmark our tagger in the presence of detector effects [46], to compare its performance to the classic quark–gluon variables, and to study the correlation with the jet momentum.

Finally, we will establish realistic and relevant benchmark analysis for quark–gluon tagging at the LHC. Unfortunately, it is already known that quark–gluon tagging does not significantly improve weak-boson-fusion analyses at the LHC [57]. Two often-discussed candidate analyses for quark-gluon tagging in LHC searches are

1. mono-jet dark matter searches with a gluon-dominated signal, Sec. 3, and

2. di-jet resonance searches with a quark-dominated signal, Sec. 4.

For both cases we motivate the use of quark–gluon tagging, show how our LoLa tagger helps extract the signal, and discuss the limitations in a realistic analysis setup.

## 2 Ideal world

In spite of the fact that a parton-level definition of quark and gluons becomes ambiguous beyond leading-order QCD, we start with an analysis of jets coming from hard quarks and gluons at tree-level and based on Monte Carlo truth. The impact of this simplification should eventually be tested including higher-order effects. At this point it will allow us to identify

the leading subjet properties of such jets and to compare our deep learning approach with established approaches.

We generate quark and gluon jet samples using di-jet events with SHERPA2.2.1 [59] at 14 TeV. We do not simulate any multiple interactions and any effects from pile up could be dealt with by using established techniques as well as recently proposed tools [60–63]. For quark jets we extract the subprocesses $gg/q\bar{q} \to q\bar{q}$ and $qq \to qq$, for the gluon jets we keep the subprocesses $gg/q\bar{q} \to gg$. We pass these events through DELPHES3.3.2 [64], using the standard ATLAS card. Finally, we cluster the particle flow objects [65] into anti-$k_T$ [66] jets of radius $R = 0.4$ using FASTJET3.1.3 [67, 68]. All jet constituents have to be central in the detector, with $|\eta| < 2.5$ and $p_T > 1$ GeV. Unless explicitly mentioned, our jets have

$$p_{T,j} = 200 \ldots 220 \text{ GeV} . \tag{1}$$

This setup closely follows Ref. [45], with an additional fast detector simulation. We do, however, find that switching from PYTHIA to SHERPA makes quark–gluon discrimination generally a little harder [42].

## 2.1 Standard observables

Distinguishing quark jets from gluon jets exploits two features [69]: first, radiating a gluon off a hard gluon versus off a hard quark comes with a ratio of color factors $C_A/C_F = 9/4$. This leads to a higher particle multiplicity ($n_{\text{PF}}$) and a broader radiation distribution or girth ($w_{\text{PF}}$) [70, 71] for hard gluons; second, the splitting functions $\hat{P}_{gg}(z)$ and $\hat{P}_{qq}(z)$ differ in the soft limits. The harder fragmentation for quarks makes quark jet constituents carry a larger average fraction of the jet energy, tracked by the variable $p_T D$ [34]. In addition, the two-point energy correlator $C_{0.2}$ separates quarks and gluons with an optimized power of $\Delta R_{ij}$ [72]. This allows us to define the four established observables

$$n_{\text{PF}} = \sum_i 1 \qquad\qquad w_{\text{PF}} = \frac{\sum_i p_{T,i} \Delta R_{i,\text{jet}}}{\sum_i p_{T,i}}$$

$$p_T D = \frac{\sqrt{\sum_i p_{T,i}^2}}{\sum_i p_{T,i}} \qquad\qquad C_{0.2} = \frac{\sum_{ij} E_{T,i} E_{T,j} (\Delta R_{ij})^{0.2}}{\sum_i E_{T,i}^2} . \tag{2}$$

In addition, we evaluate the highest fraction of $p_{T,\text{jet}}$ contained in a single jet constituent [73], and the minimum number of constituents which contain 95% of $p_{T,\text{jet}}$ [74],

$$x_{\max} \qquad \text{and} \qquad N_{95} . \tag{3}$$

The latter is obviously correlated with the number of constituents $n_{\text{PF}}$. All jet constituents summed over are defined as Delphes E-flow objects, combining both the calorimeter and the tracking information.

Distributions of all these observables for pure quark and gluon samples are shown in Fig. 1, both in an ideal setup and at the level of particle flow object after fast detector simulation. The IR-sensitive and theoretically challenging observable $n_{\text{PF}}$ shows large differences because LHC detectors rapidly lose sensitivity for soft constituents. The $p_T D$ distribution is similarly sensitive. When we add a soft constituent we find that the numerator and denominator change differently,

$$p_T D \sim \frac{\sqrt{p_T^2 + \epsilon^2 p_T^2}}{p_T + \epsilon p_T} \approx \frac{1 + \epsilon^2/2}{1 + \epsilon} . \tag{4}$$

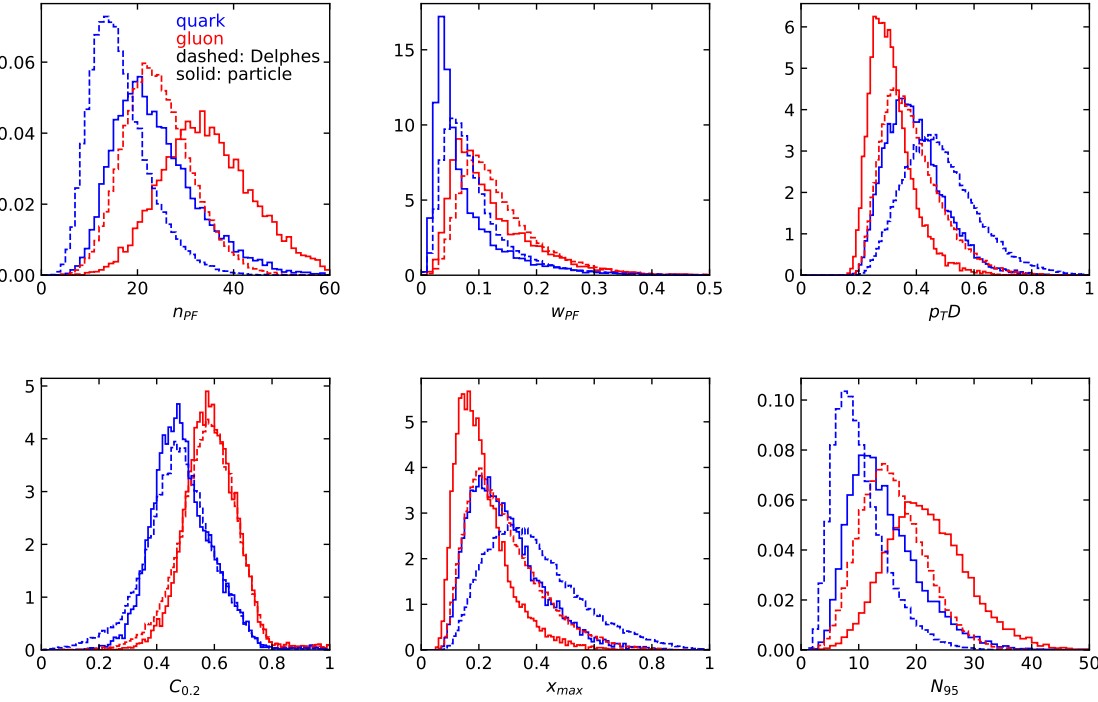

Figure 1: Normalized distributions for the subjet variables described in the text for pure quark and pure gluon jets, with and without detector effects. Jets are selected with $p_T = 200 \ldots 220$ GeV.

This way $p_T D$ shifts towards smaller values, which do not survive a detector simulations, as seen in Fig. 1. The situation is more stable for the $p_T$-weighted $w_{\mathrm{PF}}$ and for $C_{0.2}$.

The individual performance of these six observables in tagging pure quark and gluon jets without detector effects is illustrated in the left panel of Fig. 2. Each of the observables indeed contributes to quark-gluon discrimination. The number of constituents $n_{\mathrm{PF}}$ is the most powerful single variable, with almost identical performance to $N_{95}$. This confirms the findings of Ref. [45] in the absence of detector effects. To maximize their separation power we combine all six of them into a boosted decision tree (BDT), implemented in SCIKIT-LEARN using a gradient boosting classifier with 50 estimators, a maximum tree depth of 4, a sub-sampling fraction of 0.9 and a learning rate of 1. The classifier is trained on a sample of 500k quark and gluon jets, 5% of which are set aside as a test sample. The corresponding ROC curves are also shown in Fig. 2, showing a small improvement over the most powerful, but poorly defined variable $n_{\mathrm{PF}}$. In the right panel of Fig. 2 we compare the ROC curves with and without detector simulation. From Fig. 1 we know that for all variables the detector affects the quark and gluon distributions systematically, both shifting and broadening the features. We can quantify the detector effect for instance by comparing the gluon tagging efficiencies with and without DELPHES as a function of the quark efficiency in the right panel of Fig. 2. The result suffers from numerical fluctuations for extremely small $\epsilon_q < 0.01$, but for the bulk of the ROC curves for each observable the detector effect are within 10% of the ideal curve. Interestingly, the simplest observables $n_{\mathrm{PF}}$ and $N_{95}$ turn out the most stable in distinguishing quarks from gluons. This suggest that they offer sizeable quark-gluon separation power already in phase space regions which are not affected by detector effects.

Given that our six jet observables are an ad-hoc collection and do not form any kind of basis in a space of correlators, it is neither guaranteed that they include all available information

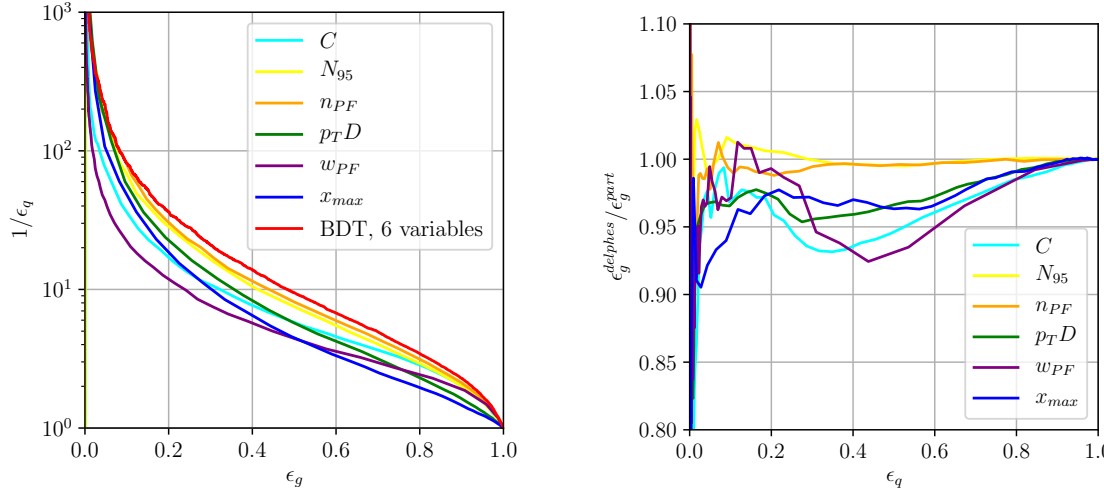

Figure 2: Left: ROC curves for the six quark–gluon observables discussed in the text, including a combination through a BDT, without detector effects. Right: detector effect illustrated as ratios of single-observable ROC curves, shown as the ratio $\epsilon_g^{\text{DELPHES}}/\epsilon_g^{\text{particles}}$.

nor that they form a minimal set. The first question can be answered when we eventually compare their separation power to our deep-learning tagger. To tackle the second question we plot the feature importance of each input variable in Fig. 3. For a variable $x$ we want to look at individual nodes $t$ making up a tree and how often $x$ is used for a split $s_t$. For each split we first compute the probability $p(t)$ for a sample to reach the node $t$ and define the purity of each node by the Gini index

$$i(t) = 1 - \sum_{\text{outcomes } j} p_j^2(t) = 2p_1(t)p_2(t) < \frac{1}{2}\,, \tag{5}$$

where the last step holds for two classification hypotheses and gives twice the probability of choosing a data point of category $j$ at node $t$, multiplied by the probability of mis-labeling it. It reaches its maximum for even probabilities and tends to zero if all the samples in a node are of the same category. In that sense it is a measure of the purity or impurity of the sample at node $t$. Next, we compute the change in purity of the node $t$ when we define a split $s_t$ in terms of the variable $x$, defining $\Delta i(s_t, t)$. This allows us to quantify the importance of a variable $x$ as

$$\text{Imp}(x) \propto \sum_{\text{trees}} \sum_{\text{nodes}} p(t)\Delta i(t,x)\,, \tag{6}$$

modulo a normalization constant. A decision tree is essentially a series of nodes which splits the samples such that the decrease in impurity is maximized, hence more important features are more often used to split the samples. Because cutting on a one-dimensional distribution as shown in Fig. 2 masks correlations, the importance allows us to define the feature that best separates the data whilst being least correlated with other variables. We show the results in Fig. 3 and find a start constrast to the single-variable results of Fig. 2. The most powerful single observables $n_{\text{PF}}$ and $N_{95}$ are strongly correlated with the leading variable $w_{\text{PF}}$ and therefore contribute little to the multi-variate analysis. Instead, the two-point correlation $C_{0.2}$, which carries extra information than the other (first-order moment) variables, is the most important additional feature. Amusingly, these two leading observables $w_{\text{PF}}$ and $C_{0.2}$ are also

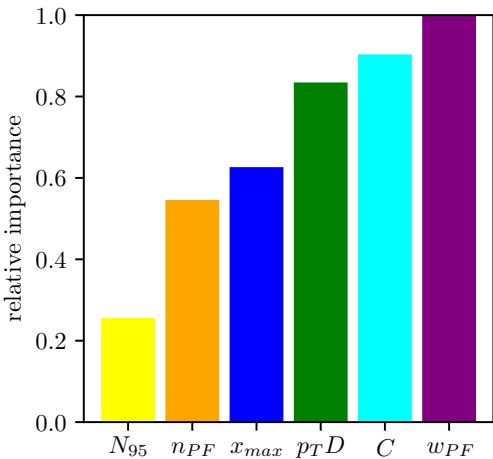

Figure 3: Feature importance of each variable in the BDT, after a DELPHES simulation, normalized to the most important feature.

IR-safe [72]. All other observables constribute to the quark-gluon separation, but with different impact.

We close with a word of caution. The subjet observables given in Eq.(2) are not theoretically well-defined observables which we can compute based on QCD. Instead, they are statistical descriptions of jet constituents, including two-object correlators, in some cases IR-modified by an appropriate energy scaling. Relying on not consistently IR-safe observables complicates quark-gluon separation at the LHC, but does not make it impossible [35–42, 44]. The main problem is that we cannot define quark or gluon jets in perturbative QCD or in Monte-Carlo simulations beyond leading order in QCD. Clearly, these observables as well as low-level observables cannot be directly used to study QCD properties of subjets. On the other hand. IR-safety does not have to be an issue for data-to-data analyses, like quark-gluon tagging trained on observed jets. All we need to do is define the quark and gluon labels in relation to a hard process which predicts mostly quarks or mostly gluons, rather than jet by jet [43]. This way we can use the potentially powerful soft and collinear subjet information as long as we do not attempt to interpret these measurements in terms of QCD.

## 2.2 Charging LoLa

Given our result for the multi-variate analysis of high-level substructure variables, it is natural to ask what happens when we attempt to capture all available information from low-level observables using a deep neural network. To combine information from the calorimeter and the tracker with its different resolution, a promising approach is the LOLA architecture applied to particle flow objects, developed for the DEEPTOPLOLA tagger [16]. The input to the network are the $N$ jet constituent 4-vectors sorted by $p_T$,

$$
(k_{\mu,i}) = \begin{pmatrix} k_{0,1} & k_{0,2} & \cdots & k_{0,N} \\ k_{1,1} & k_{1,2} & \cdots & k_{1,N} \\ k_{2,1} & k_{2,2} & \cdots & k_{2,N} \\ k_{3,1} & k_{3,2} & \cdots & k_{3,N} \end{pmatrix} .
\tag{7}
$$

Since $N$ varies from jet to jet, we zero-pad jets with fewer than $N$ constituents, and increase $N$ until the tagging performance is saturated, for most physics scenarios giving $N = 25 \ldots 30$. Above this the soft jet constituents carry too little information to compensate for the increasing

computation time. Inspired by the structure of recombination jet algorithms, we multiply the original 4-vectors with a trainable matrix $C_{ij}$, defining a combination layer (CoLa)

$$k_{\mu,i} \xrightarrow{\text{CoLa}} \widetilde{k}_{\mu,j} = k_{\mu,i}\, C_{ij}$$

$$\text{with} \quad C = \begin{pmatrix} 1 & 1 & \cdots & 0 & \chi_1 & \cdots & 0 & C_{1,N+2} & \cdots & C_{1,M} \\ \vdots & & \ddots & & & \ddots & & \vdots & \ddots & \vdots \\ 1 & 0 & \cdots & 1 & 0 & \cdots & \chi_N & C_{N,N+2} & \cdots & C_{N,M} \end{pmatrix}. \tag{8}$$

This increases the number of inputs from $N$ to $M$, where $M$ is a tunable hyper-parameter of the network. The entry $\chi_j$ is new for the quark–gluon implementation and encodes the information if a particles is charged or not, $\chi_j = 0, 1$ [45]. For most of the phase space considered in this paper, we will find that the tagging performance for our specific applications hardly improves, but obviously this result should not be generalized. To make it easier for the network to learn the mathematical structure of Lorentz transformations we pass the CoLa output to a Lorentz layer (LoLa)

$$\tilde{k}_j \xrightarrow{\text{LoLa}} \hat{k}_j = \begin{pmatrix} [c]m^2(\tilde{k}_j) \\ p_T(\tilde{k}_j) \\ p_T(\tilde{k}_j)\Delta R_{j,\text{jet}} \\ w_{jm}^{(E)} E(\tilde{k}_m) \\ w_{jm}^{(d)} d_{jm}^2 \\ E_T(\tilde{k}_j)E_T(\tilde{k}_m)(\Delta R_{jm})^{0.2} \end{pmatrix}, \tag{9}$$

with $d_{jm}^2 = (\tilde{k}_j - \tilde{k}_m)^2$. To adapt this layer to quark–gluon separation we augment it with the third and the last entries. They follow the definition of the the subjet variables $w_{\text{PF}}$ and $C$ in Eq.(2), with the sum over constituents stripped off so that they are defined per constituent. The first three $\hat{k}_j$ map individual 4-momenta $\tilde{k}_j$ onto their invariant mass and transverse momentum. The fourth entry is a linear combination of all energies with trainable weights $w_{jm}^{(E)}$, while the fifth entry sums over the Minkowski distance between $\tilde{k}_j$ and all other 4-momenta $\tilde{k}_m$, again weighted by $w_{jm}^{(d)}$ which is updated after each training epoch. For the lower three entries we can either sum over or minimize over $m$ while keeping $j$ fixed. For $w_{jm}^{(E)}$ we choose the sum over the internal index; for $w_{jm}^{(d)}$ we include four copies with independently trainable weights, two summing and two minimizing over the internal index; for the last entry we use two copies, one with a sum and one with a minimum. However, it turns out that the new LoLa observables have limited impact on the quark–gluon separation, independent of the options applied to the last the last entry in Eq.(9).

After the LoLa stage, the inputs are passed through ReLU-activated dense layers with 100 and 50 units and dropout rate 0.2 and 0.1, respectively. Both dense layers have an additional L2 regularization of $5 \times 10^{-4}$ and are initialized with He-normal functions. A final dense layer converts the weights into a normalized score with SoftMax activation. All training is done using KERAS [75] with the THEANO [76] back-end on a GPU cluster. The hyper-parameters are optimized with ADAM [77], using a learning rate of $10^{-5}$ and a batch size of 128. We have checked that both, for the size of the training sample and for the number of constituents our performance reaches safe plateaus. Throughout this paper we use $N = 80$ constituents, significantly above where we would expect the soft activity to be universal.

Turning to the performance, we plot the ROC curves for our best-performing LoLa architecture in the left panel of Fig. 4, compared to the 6-observable BDT,

$$\left\{ n_{\text{PF}}, w_{\text{PF}}, p_T D, C_{0.2}, x_{\text{max}}, N_{95} \right\}. \tag{10}$$

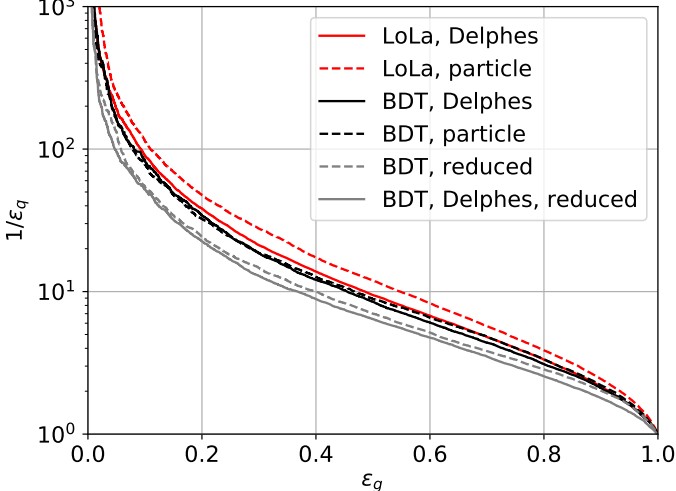
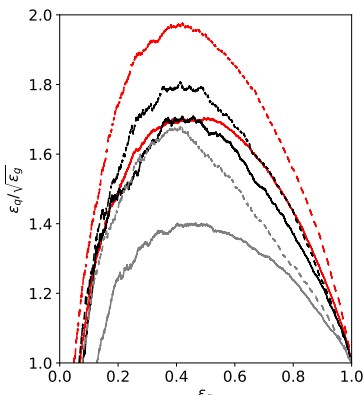

Figure 4: ROC and significance improvement curves for the LoLa tagger trained and tested on pure samples, with and without detector simulation. We compare to a BDT analysis of the full set of of observables, Eq.(10), and to a reduced set of observables, Eq.(11).

In the right panel we also show the increase in signal significance as a function of the signal efficiency, to help us optimize the impact of the tagger as an analysis tool for instance in terms of SI $= \epsilon_g / \sqrt{\epsilon_q}$. With our LoLa network we reproduce the performance of the enhanced images setup of Ref. [45] without detector simulation and after accounting for the move from PYTHIA to SHERPA. Our agreement is at the level that different trainings of our LoLa tagger on the framework of Ref. [45] shower a stronger variation than the agreement between the LoLa and the CNN performance. Different architectures without detector effects are studied in detail in Ref. [44]. They are very close in performance, including convolutional networks like that of Ref. [45], and we have good reason to assume that this pattern will not change once we include detector effects.

We also note an overall improvement with respect to our 6-observable BDT. The fact that the deep network does not hugely outperform the multi-variate analysis on the subjet level is not unexpected. The difference between the LoLa network and the BDT becomes smaller once we include detector effects. This points to the deep network finding additional information which even the theoretically poorly defined observables do not capture. As a test of stability we also show BDT results with a reduced and less IR-sensitive set of observables,

$$\left\{ p_T D, C_{0.2}, x_{\max}, N_{95} \right\} . \tag{11}$$

As we can see in Fig. 4 this reduces the over-all performance of the BDT, but does not improve the stability with respect to detector effects.

Finally, we need to test if the quark–gluon network correctly captures the information we know exists at the subjet level [78, 79]. Because we have access to Monte Carlo truth we can, for instance, plot the distributions of our six observables for quark jets identified as quarks and for gluon jets identified as gluons. We can compare these distributions between the LoLa network, the BDT, and the truth information, all including detector effects. In Fig. 5 we plot all observables introduced in Sec. 2.1, at truth-level and after selecting the 30% best-identified jets. For gluon jets the classifier favors slightly lower values of $p_T D$ and $x_{\max}$, and larger values of $C$, $N_{95}$ and $n_{\mathrm{PF}}$. A significant sculpting of these distributions relative to truth indicates a challenge in separating the two hypotheses. The observables where LoLa best matches the

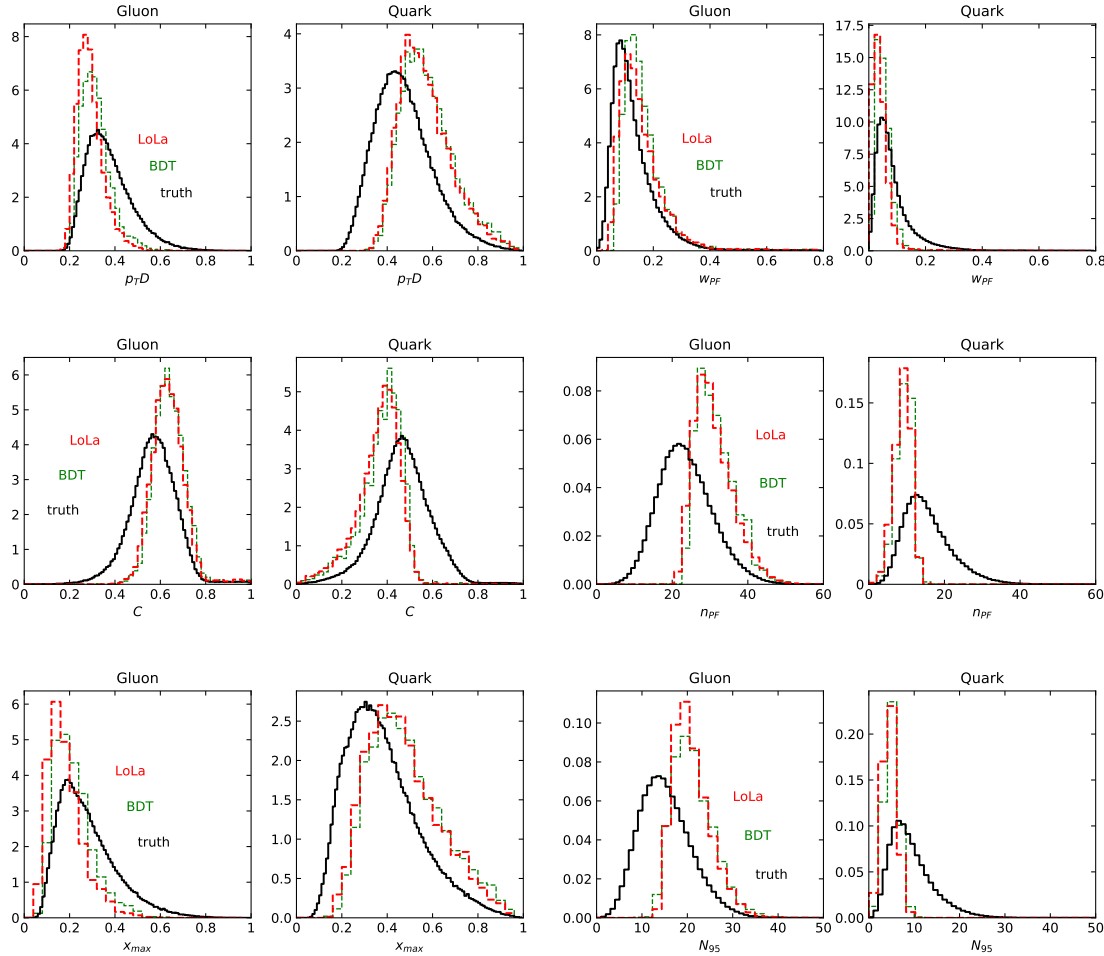

Figure 5: Distributions for the sub-jet observables. The black curves show the truth from Fig. 1. The red dotted curves are the 30% most gluon-like or quark-like jets from LoLa, the green curves from the BDT.

truth are $w_{\text{PF}}$ and $C_{0.2}$. These are also the two most important observables in the BDT in Fig. 2, indicating that the BDT and LoLa indeed rely on similar information.

Table 1: Areas under the ROC curve for the LoLa tagger trained and tested on pure samples sliced in $p_{T,j}$. The uncertainty on each entry is one to two units on the last shown digit.

| Train | Test | | | | |
|---|---|---|---|---|---|
| | 200-210 | 210-220 | 220-230 | 230-240 | 240-250 |
| 200-210 | 0.812 | 0.812 | 0.812 | 0.818 | 0.816 |
| 210-220 | 0.812 | 0.813 | 0.812 | 0.819 | 0.817 |
| 220-230 | 0.804 | 0.805 | 0.810 | 0.811 | 0.808 |
| 230-240 | 0.803 | 0.804 | 0.801 | 0.814 | 0.809 |
| 240-250 | 0.810 | 0.811 | 0.811 | 0.820 | 0.818 |

| Train | Test | | | | |
|---|---|---|---|---|---|
| | 200-250 | 300-350 | 400-450 | 500 - 550 | 600-650 |
| 200-250 | 0.813 | 0.818 | 0.805 | 0.782 | 0.74 |
| 300-350 | 0.811 | 0.825 | 0.823 | 0.818 | 0.80 |
| 400-450 | 0.809 | 0.824 | 0.834 | 0.838 | 0.80 |
| 600-650 | 0.807 | 0.816 | 0.830 | 0.840 | 0.841 |

## 2.3 Jet kinematics

One dangerous sources of systematic uncertainties in subjet physics and elsewhere is mismeasuring the momentum of the jet [80]. Because the structure of parton splittings is sensitive to the range of energies described by the splitting history, we do not want to remove this information for example through an adversarial network. Instead, we want to include $p_{T,j}$ in the information available to the tagger. Before we do so, we need to understand at what level the quark–gluon network is sensitive to the transverse momentum of the jet [45, 46].

To this end we train and test individual LoLa networks in different slices of $p_{T,j}$, again with detector effects, and test them on over a range of transverse momenta. We show the AUC values for different combinations of training and testing samples in Tab. 1. The left table shows the performance of the network for a small step size $\Delta p_T = 10$ GeV. On the diagonal we see that the performance of the network slightly increases towards higher momenta. This can be understood through the larger number of constituents radiated off initial partons with higher momentum. For the off-diagonal entries there is also a small generic trend that using a network on somewhat higher-$p_T$ jets than it was trained for does not reduce its efficiency. Because the differences between quarks and gluons are more subtle for softer jets, a network trained on these subtle differences may also be applied to harder jets. However, in the other direction the network trained on the more obvious hard jets will slightly deteriorate for softer jets. In the right table we test a wider range of transverse momenta. We observe the same trend, but for networks trained between 200 and 350 GeV the performance seriously suffers when we compare it to $p_T > 600$ GeV.

We only show central values in both of these tables, but we have estimated uncertainties on the performance measures in two ways. The larger error bar comes from using a trained network on different test samples, it gives typical uncertainties of $\Delta$AUC $\approx 0.002$ for most of the entries, increasing to $\Delta$AUC $\approx 0.01$ for the larger separations in $p_T$. The error we find from using different trainings on the same test sample is, in our case, about an order of magnitude smaller.

For the $p_{T,j}$ slices in Tab. 1 we can compute the ROC curves for the LoLa quark–gluon discrimination. In the left panel of Fig. 6 we see how the performance of the tagger is stable, with a slight increase in performance towards higher jet momenta.

In the right panel of Fig. 6 we repeat the same exercise, but including the charge information discussed in Eq.(8). Indeed, the performance is unchanged for this specific change in the LoLa setup, at least up to $p_{T,j} < 600$ GeV and once we include detector effects.

## 3   Mono-jets

To see at what level quark–gluon discrimination really helps at the LHC we need benchmark applications. For WBF jets we have unfortunately seen that the substructure of the tagging jets can alleviate the pressure on global observables like a central jet veto, but that the signal vs background system is already over-constrained by event-level kinematic information and jet substructure [57]. We therefore turn to the simplest jet analyses with the fewest number of established handles to control the background.

Our first candidate is the mono-jet signature probing invisible decays of a SM-like Higgs boson. Here, the transverse momentum of the tagging jet is essentially the only kinematic variable used in standard analyses. Far from the expected performance of the leading WBF and $VH$ channels for invisible SM-like Higgs decays, this mono-jet channel is extremely versatile when we search for dark matter or want to learn more about the nature of an invisible Higgs signal. For a Higgs-like mediator it provides us with a benchmark process for a tagger extracting a gluon-dominated signal from a quark-dominated background [81, 82]. Obviously, all our

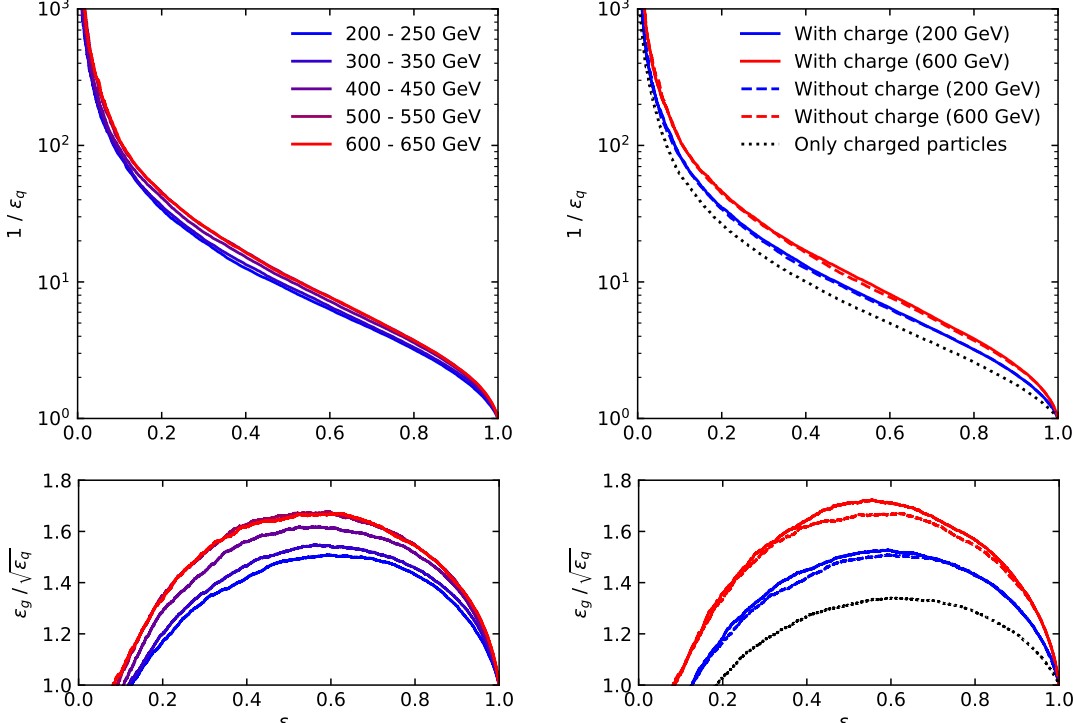

Figure 6: Left: ROC and SI curves for the pure quark and gluon samples in non-overlapping jet $p_T$ ranges. Right: ROC and SI curves for the pure quark gluon samples including charge information.

findings can be generalized to searches for (pseudo-)scalar mediators at the LHC. For those the relative importance of the electroweak WBF and $VH$ channels compared to the gluon-induced mono-jet channel can obviously be completely altered.

The key feature of mono-jet searches with scalar mediators is that the signal jet is almost always gluon-initiated, while for the $Z$+jets background it is mostly quark-initiated, as illustrated in Fig. 7. Increasing $p_{Tj}$ pushes the events kinematics towards larger proton momentum fractions and enhances the quark contribution, slowly reducing the gluon purity of the Higgs signal. Observing such a signal in mono-jet events requires exquisite control of the large backgrounds from $V$+jets production. While the largest background is $Z(\to \nu\nu)$+jets, there exists a sizeable irreducible contribution from $W(\to l\nu)$+jets, where the lepton either fakes a jet or escapes undetected [83]. Due to the rather inclusive signature of a high-$p_T$ jet with large missing transverse energy, there is little to cut on other than either $p_{T,j}$ of $\not{E}_T$. In practice, a cut of at least $\not{E}_T \geq 100$ GeV is typically required at the trigger level.

We generate the $H$+jets signal events, including the finite top mass effects with SHERPA2.2.1 [59] and OPENLOOPS [84] at a collider energy of 14 TeV. For the $Z$+jets background we also use SHERPA2.2.1 [59] with the COMIX for matrix element generation [85], and we employ CKKW-L merging [86–88] with up to two jets in the matrix element for both $H$+jets and $Z$+jets. As in the case of the pure samples, we use $\Delta R = 0.4$ anti-$k_T$ jets with all visible final-state particles of $|\eta| < 2.5$ as constituents [67, 68]. As long as we stick to leading-order simulation we can extract the parton content for example of the hardest jet from Monte Carlo truth.

To illustrate the challenge in observing this signal, we plot some kinematic distributions for the signal and background in Fig. 8. Note that following the discussion in Sec. 2.1 we do not distinguish gluon jets from quark jets, but the Higgs plus jets signal from the $Z$ plus jets background. First, the expected signal-to-background ratio even assuming an invisible Higgs branching ratio of formally 100% is at the per-mille level. Second, the leading jet kinematics

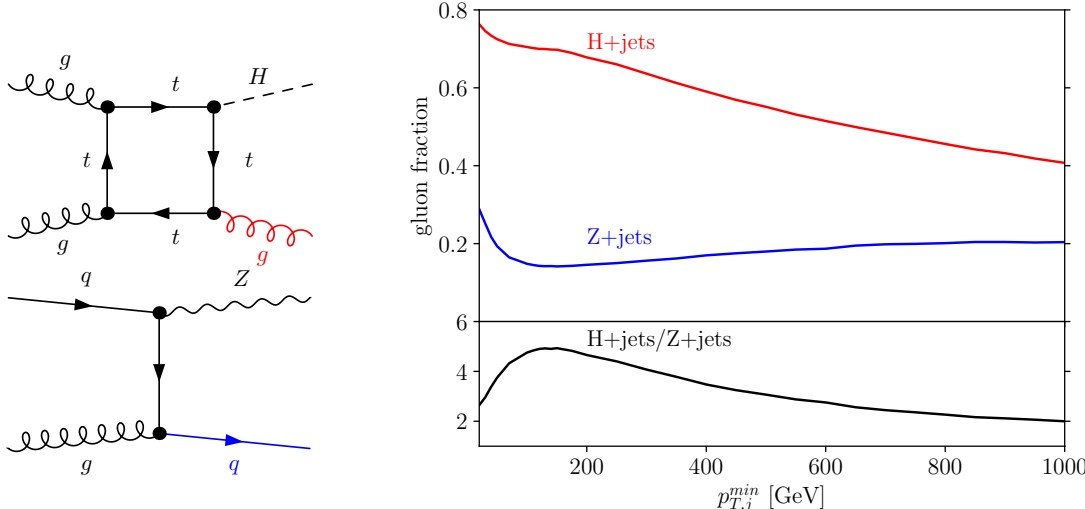

Figure 7: Left: representative Feynman diagrams for the mono-jet signal and $Z$+jets background. Right: fraction of events with a leading jet above $p_{T,j}^{\min}$ with gluon-initiated leading jets in $H$+jets events (red) and quark-initiated jets in $Z$+jets events (blue) as a function of $p_{T,j}$. The bottom panel shows the ratio of the two gluon fractions.

for the signal and background is essentially identical, while the second jet is actually softer in the signal. A cut-and-count analysis above a stringent $\not{E}_T$ requirement is not an optimal analysis strategy, because the small difference between the Higgs and $Z$ masses hardly affects the kinematics. Of course, if the mono-jet signal is due to a light mediator, the signal $p_T$-spectrum will be harder.

A subjet feature, which is not exploited in the event-level analysis is that the hardest background jet is quark-initiated in 80% of events, while the leading signal jet is usually gluon-initiated. From Fig. 7 we expect the quark–gluon tagger to be most useful at low to intermediate $p_{Tj}$. To study this question quantitatively, we generate mono-jet samples in non-overlapping slices of $p_{T,j}$ and train and test LoLa on all combinations of the above samples. The performance of each combination, given by the area under the curve (AUC), is shown in Tab. 2. These numbers can be directly compared to their counterparts for pure samples in Tab. 1. We see that the diagonal entries, corresponding to networks trained and tested in the same $p_T$ range, show the best performance, and the performance gradually decreasing with $p_T$, reflecting the drop in quark vs gluon purity shown in Fig. 7.

Table 2: Areas under the ROC curve for the LoLa tagger trained and tested on mono-jet samples sliced in $p_{T,j}$. The uncertainty on each entry is one to two units on the last shown digit.

| Train | Test | | | | |
|---|---|---|---|---|---|
| | 200-210 | 210-220 | 220-230 | 230-240 | 240-250 |
| 200-210 | 0.692 | 0.692 | 0.691 | 0.692 | 0.687 |
| 210-220 | 0.692 | 0.692 | 0.692 | 0.692 | 0.687 |
| 220-230 | 0.692 | 0.692 | 0.692 | 0.692 | 0.688 |
| 230-240 | 0.692 | 0.692 | 0.692 | 0.692 | 0.688 |
| 240-250 | 0.692 | 0.692 | 0.692 | 0.692 | 0.688 |

| Train | Test | | | |
|---|---|---|---|---|
| | 200-250 | 250-300 | 300-350 | 600-650 |
| 200-250 | 0.691 | 0.683 | 0.674 | 0.604 |
| 250-300 | 0.691 | 0.685 | 0.677 | 0.605 |
| 300-350 | 0.687 | 0.683 | 0.677 | 0.614 |
| 600-650 | 0.630 | 0.638 | 0.646 | 0.631 |

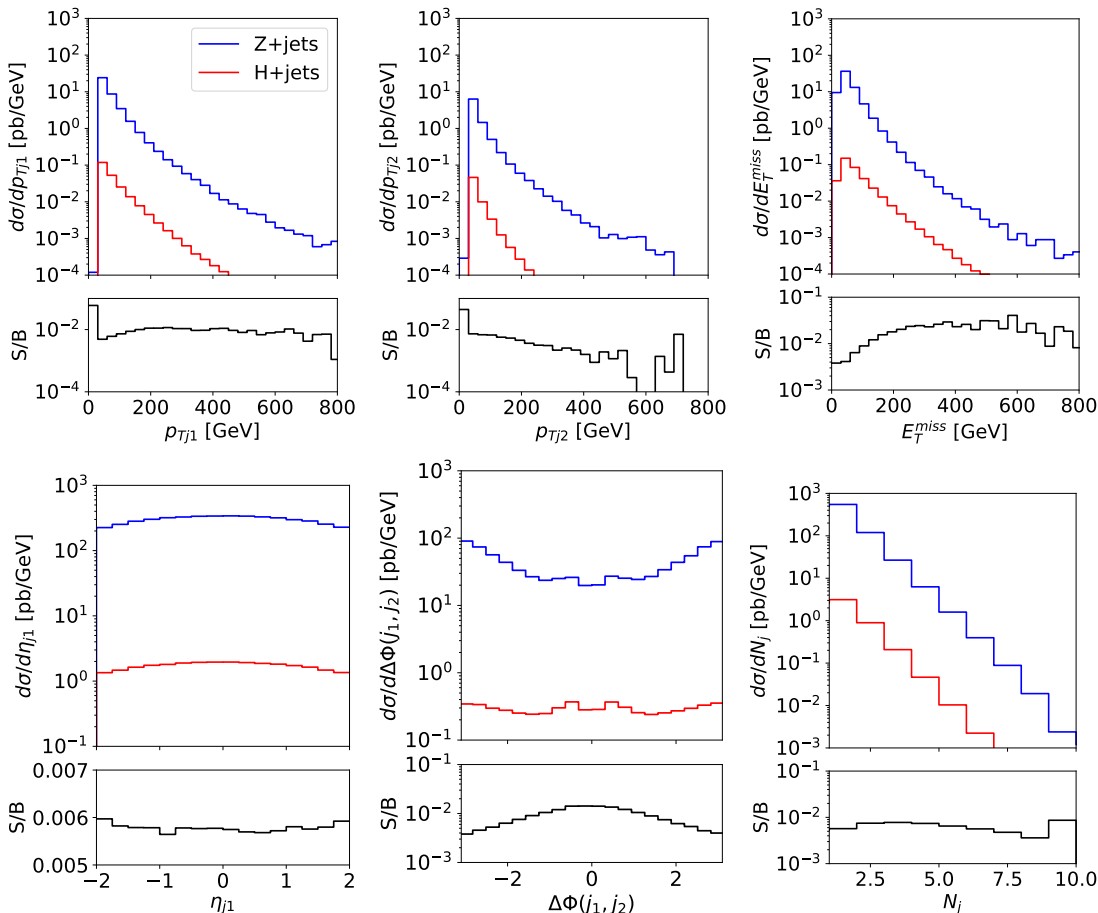

Figure 8: Kinematic distributions for the $H$+jets signal (red) and leading $Z$+jets background (black), along with the signal-to-background ratio. We show the leading $p_{T,j}$, the second $p_{T,j}$, $\not{E}_T$, the pseudorapidity of the leading jet, $\Delta\phi$ of the leading two jet, and the jet multiplicity.

The ROC curves corresponding to the diagonal train and test combinations of Tab. 2, and their corresponding SI curves, are shown in Fig. 9. All curves show the same behavior, with the drop in performance for high-$p_T$ jets visible for the 600 ... 650 GeV slice. For the actual mono-jet analysis this implies that quark–gluon discrimination is least efficient when the analysis focuses on the kinematic regime with the largest missing energy. However, from Fig. 8 we know that for heavy mediators like a SM-like Higgs this kinematic range is not the most promising. Instead, we typically analyze the entire $p_{T,j}$ distribution and extract a signal significance from a shape analysis in the presence of large systematic uncertainties. This is the reason why we cannot quote a simple significance improvement for the mono-jet analysis from quark–gluon tagging. Also for lighter mediators, the bulk of the $\not{E}_T$ distribution is what allows us to control the backgrounds at the required level [83], and here a systematic application of quark–gluon tagging may improve our limited event-level understanding of signal vs background features. On the other hand, at this level it should be clear that for quark-gluon discrimination in the presence of detector effects the mono-jet channel does not provide a useful benchmark.

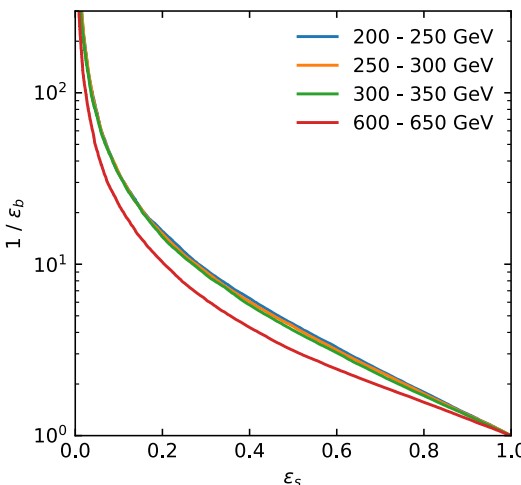

Figure 9: ROC curves for the mono-jet samples in non-overlapping jet $p_T$ ranges.

# 4 Di-jet resonances

As a second application, we study resonances decaying to two jets. These signal decay jets are usually quark-initiated, while for relatively light resonances the background will be multi-gluon production. An interesting aspect of this analysis is that we could, in principle, use this quark–gluon information already at the trigger level to enhance the LHC reach in di-jet resonance searches.

We consider an axial vector $Z'$ with a democratic coupling to all quarks, ignoring the obvious problems with a UV completion [89–94]. This resonance might or might not be a dark matter mediator — in this study we only consider its decay to quarks described by the Lagrangian [95]

$$\mathcal{L}_{Z'} = g_{Z'} \sum_q Z'_\mu \overline{q} \gamma^\mu \gamma_5 q + \cdots . \tag{12}$$

The decay to quarks has the benefit that the entire signal only depends on one kind of coupling, and exactly the coupling we eventually need to quantitatively analyze mono-jet signals when the new resonance is a dark matter mediator. We consider two benchmark point for the $Z'$ mass, namely $m_{Z'} = 450$ GeV and $m_{Z'} = 750$ GeV, combined with $g_{Z'} = 0.1$, and simulate the signal and the background with SHERPA2.2.1 [59] to leading order. The selection criteria for a standard LHC search are at least two jets with [96]

$$p_{T,j_1} > 220(185) \text{ GeV} \qquad p_{T,j_2} > 85 \text{ GeV} \qquad |\eta_j| < 2.8 , \tag{13}$$

combined with the resonance-inspired requirements

$$|y^*| = \frac{|y_{j_1} - y_{j_2}|}{2} < 0.6(0.3) \quad \text{and} \quad \frac{p_{T,j_1} + p_{T,j_2}}{2} = (0.6 \dots 1.4) p_{T,j_1} . \tag{14}$$

In the left panel of Fig. 10 we first analyze the leading jet for the low-mass case and both jets for the heavy-mass case. In both cases we use the pre-trained networks from the pure samples. We find that the quark–gluon tagging works slightly better for lower-mass resonances or lower typical $p_{T,j}$. This has nothing to do with the signal and is driven by the purity of the QCD background in this phase space region. The second jet from the light resonance is comparably soft, which makes it hard to separate it from QCD radiation without strongly shaping the background.

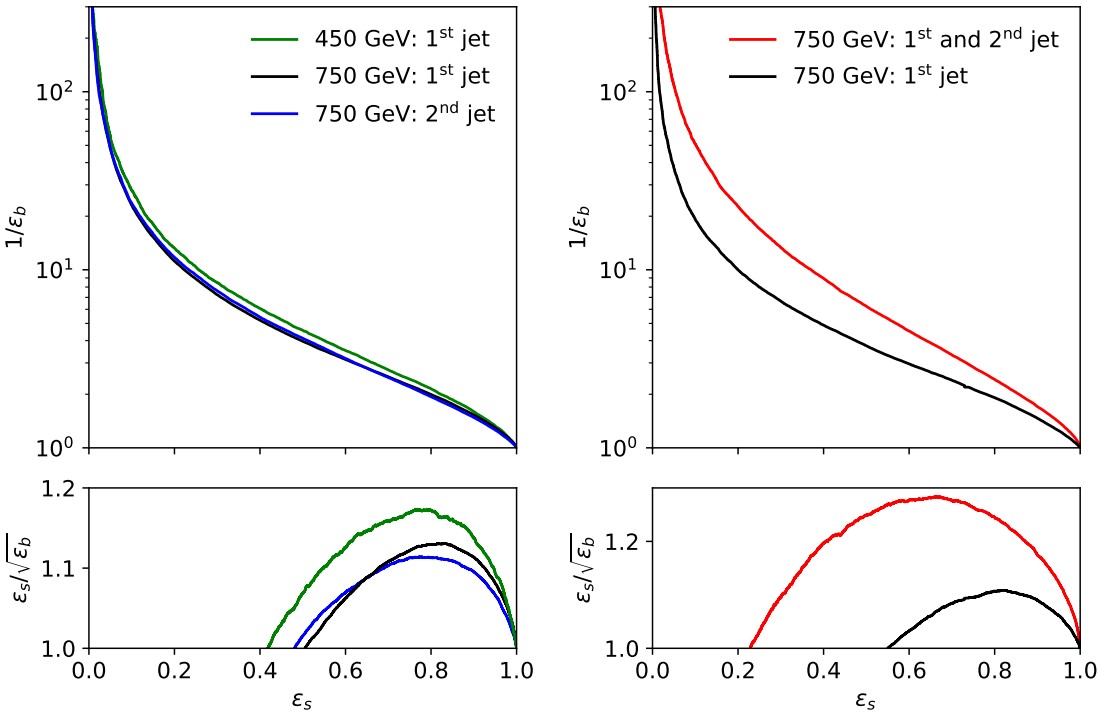

Figure 10: Left: ROC and SI curves for a 250 ... 300 GeV jet from a 450 GeV $Z'$ resonance and for 300 ... 400 GeV jets from a 750 GeV $Z'$ resonance, using the model trained on pure samples. Right: performance improvement from considering both decay jets from a 750 GeV $Z'$ resonance, based on a dedicated training.

We also see that, for $m_{Z'} = 750$ GeV the harder jet has more sensitivity for high signal efficiencies, whereas the second hardest jet has more sensitivity for lower signal efficiencies. Consequently, in the right panel of Fig. 10 we show the performance of a dedicated two-jet LoLa network, combining the network output from the two jets into an additional set of layers and then producing the standard di-jet tagging output. As expected, the signal and the background independently predict two quarks and two gluons, so the combined network efficiency receives a significant boost. On the other hand, it is well known that there exist a wealth of observables which are sensitive to the quark vs gluon nature of jets at the event level, like additional jet activity. This kind of information is fully correlated with the quark-gluon tagging of the di-jets, and it is unlikely that the jet tagging significantly improves the LHC reach once all event-level observables are considered [57]. On the other hand, these event-level observables are non-trivial to control, so adding quark-gluon tagging should help controlling the backgrounds. In that sense, just as for the mono-jet case, our simple significance estimate is not the whole story. Resonance searches are only partly limited by statistical significance. Enriching the signal samples with quarks at an early stage will generally suppress multi-jet backgrounds. Because trained neural networks are fast, they could be used already at the trigger level to provide an improved event sample and to allow for searches in tough phase space regions.

# 5   Summary

Quark–gluon separation is one of the hardest problems in contemporary LHC physics. Technically, is has received a huge boots from machine learning on low-level observables. Also on the theory side, the general move towards likelihood-free analyses just comparing fully simulated and observed events at the detector level circumvents some of the fundamental QCD problems. In combination, these developments call for a realistic study of these methods using benchmark signal processes.

We have extended our LOLA tagger, previously used for top tagging, to statistically separate quarks from gluons. For the ideal case of pure quark and gluon jets we find that detector effects lead to a degradation of the machine learning results, to a point where a classic BDT analysis becomes competitive. However, we also remind ourselves that the standard observables entering the BDT are neither theoretically nor experimentally preferable and also show non-trivial correlations. Including charge information in LOLA can be useful for hard jets. Finally, we have shown that training and testing the network on sliced of $p_{T,j}$ leads to surprisingly stable results.

Our first benchmark channel is mono-jet production with a gluon-rich signal. Subjet information can be added to an otherwise very limited number of event-level observables. It has the potential to improve the LHC reach, especially when we use it to understand and control the entire $p_{T,j}$ distribution. The impact of $p_T$-dependent training on the systematic uncertainties should be easily controllable.

The second benchmark channel are di-jet resonances with their quark-rich signal. We find that applying a network trained on pure samples already improves the reach for relatively light $Z'$ bosons just using their couplings to quarks. Using our LOLA setup we find that for hadronically decaying $Z'$ bosons with masses below the TeV range the quark–gluon discrimination can be useful.

Altogether, we have shown that quark–gluon tagging is a theoretical and experimental challenge, that deep learning provides competitive taggers, and that their tagging performance is significantly affected by detector effects. At the LHC, there exists a range of applications, both with quark-rich and gluon-rich signals, for which it would be interesting to see how quark–gluon tagging affects triggering, background systematics, or the signal extraction in a properly described experimental setup. Unfortunately, just like weak boson fusion [57] neither mono-jet searches nor di-jet resonance searches are obvious benchmarks to estimate the impact of quark-gluon tagging on the LHC reach.

# Acknowledgments

We are very grateful to Michael Russell for his contributions during an early phase of this project. We also would like thank Monica Dunford and Hanno Meyer zu Theenhausen for very useful discussions about the di-jet channel. Finally, we acknowledge support by the state of Baden-Württemberg through bwHPC and the German Research Foundation (DFG) through grant no INST 39/963-1 FUGG (bwForCluster NEMO).

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
