# Peer review of "Quark-Gluon Tagging: Machine Learning vs Detector"

_SciPost Physics, doi:SciPost Phys. 6, 069 (2019)_

## Round 1 · Referee Report · Anonymous (Referee 1) · 2019-3-15

Strengths

1-Thorough machine-learning analysis of the problem of quark/gluon discrimination applied to two search regions

2-Analysis of benchmark, standard observables which to compare with the machine learning results

Weaknesses

1-Introduction and motivation of this work conflates and confuses many different aspects of theoretical studies, such as the importance and relevance of IRC safety

2-While the comparison to standard observables is good, the corresponding re-interpretation and combination to represent the "best" high-level observable tagger is misleading, especially when comparing to LoLa

3-I think the title of the article is misleading: "Quark-Gluon Tagging: Machine Learning meets Reality" would, to most people I think, imply that machine learning is being applied to experimental data, i.e., "reality". However all analyses in the paper are on Monte Carlo simulation.

Report

The article "Quark-Gluon Tagging: Machine Learning meets Reality" applies machine learning methods, specifically the authors' LoLa algorithm, to the problem of quark versus gluon tagging in the context of two searches. I think the ultimate results of the paper are interesting and topical, but the introduction and comparison to high-level observables is a bit confusing and uses imprecise language. I would request the authors make minor changes to the article before I recommend it for publication.

Requested changes

1-First, the title should be changed. No analysis of experimental data is performed, therefore the authors are not working with "reality". Conflating simulation with data is ultimately detrimental to experimental science.

2-The authors cite their reference [26] several times throughout the paper when discussing theoretical issues of quark versus gluon discrimination. These citations include in the second and third paragraphs of the introduction and immediately before section 2.2. I am not sure why this reference is the catch-all for theory issues with quark/gluon tagging, as reference [26] consists of two papers published in 2018, and many issues were identified years or even decades earlier. I urge the authors to provide more representative references in place of [26]. For example, the first reference of [26] mentions the serious theoretical challenges of quark/gluon discrimination, but this was presented as a Les Houches report in their reference [27]. Further, the second reference of [26] mentions infrared and colliear safety, but this issue was known decades ago.

3-I'm somewhat confused by the setup in the first paragraph of the introduction regarding "kinematic observables". The authors write that there is a change from measuring kinematic observables on jets to measuring the particle four-vectors directly with machine learning. While I understand what the authors are trying to say here, a particle's momentum four-vector is the fundamental kinematic observable. The authors should clarify these statements and more precisely and distinguish between what they mean by "kinematic observable" and what they propose in this paper.

4-At the beginning of section 2, the authors make a couple of imprecise statements. In the first sentence of section 2, they write that "quarks and gluons are poorly defined in perturbative QCD". More precisely, quark and gluon flavor jets are ambiguous in perturbative QCD, and require some definition as there is no preferred definition. In the second paragraph of section 2, the authors write that "pile up could be dealt with by using standard techniques." I understand that this article is likely only to be read by other subject experts, but the authors could provide a few references to some "standard techniques" for context for researchers outside of the field. The authors should make these changes.

5-Finally, the authors should be careful with the interpretation and implications of their 6 variable BDT that is compared to LoLa. The authors demonstrate that each of the 6 observables are individually good quark versus gluon discriminants, but this in no way means that their combination is an "optimal" discriminant. It could be that the information in jets that they use for discrimination is identical, so when combined in a BDT would not improve performance. One is only guaranteed to have an optimal tagger if the observables feed to the machine form some complete basis on phase space. Individual particle four-vectors of course do this, which is why LoLa performs so well. However, the authors should add caveats in their construction and comparison to the 6 observable BDT. Locations in the draft where qualification should be added include the discussion on page 5 and page 8.

---

## Round 1 · Referee Report · Anonymous (Referee 2) · 2019-4-3

Strengths

1) The paper is clearly written. 2 Quark-gluon tagging is a topical and interesting subject which a number of different machine-learning techniques are being applied to. A more realistic appraisal of the utility of these is desirable.

Weaknesses

1 One of the main claims of the novelty of this paper is running Delphes. This should by now should be a standard in the field and not a novelty worthy of publication. Detector effects for ML studies of quark-gluon tagging have already been included in studies such as the EFN paper 1810.05165 and Cheng's study with RNN's 1711.02633.

2 Improvements from using ML in monojet and dijet analyses are not studied in depth and are unquantified.

3 I don't agree with the statements on how well-defined the QCD variables are.

Report

This paper studies the effects of detector simulation on a machine-learning based quark-gluon tagger, based on the authors' previous work (the LoLa tagger). The paper is reasonably interesting, but it would have been better had the authors taken the time to perform a more in-depth investigation into the application of ML-based quark-gluon taggers in realistic analyses. Indeed I think the authors really punt on this issue. Another issue is confusion over the status of what actually well-defined QCD variables such as girth (or whatever one wants to call it) and two-point energy correlators. Accordingly there are a number of issues below I would like to see dealt with before publication.

Requested changes

1) Change the title. Machine Learning does not meet reality in this paper.

2) The authors need to correct their statements about QCD variables. Girth and the two-point energy correlator C2 are both IRC safe variables: both can be written as angularities with κ=1, which are IRC-safe as a class - see 1305.0007 for instance. So I don't understand or agree with the statement on p.6 that 'the subjet variables are not theoretical well-defined observables which we can compute based on QCD'. There are QCD calculations of jet broadening for instance, which is basically equivalent to girth, 1401.2158. This should be changed, along with statements in the conclusions about the observables not being 'theoretically preferable' and on p.8 about 'theoretically poorly defined observables'. Writing an interesting paper about ML techniques does not require trashing analytic defined variables.

3) Does girth really first appear in [47]?

4) There is relevant previous work on q/g tagging in the monojet channel which the authors should cite, for instance Agrawal and Rentala 1312.5325, and the shower-deconstruction paper 1607.06031.

5) Is there a reason why the IRC-unsafe variables npf and N95 are the most stable under detector simulation?

6) In the trainable matrix eq. 7, it's not clear to me what the horizontal \cdots correspond in the first part of the matrix: looking at the top row of C I see "10": are the entries in between all 1 or all zero? Or are they set by the network training? This should be clarified.

5) There is a large literature on quark-gluon tagging and appropriate definitions of q/g jets which appeared before the work of Thaler et al (which the authors are very fond of citing). One should at least cite 0601139 in this context.

6) What's the relevance of refs [54] for information contained in subjets, on p9? The Oregon paper is on data-planing, which I don't see deployed here.

Typos: p.2, paragraph 1: 'jet merely define' -> jets
p.2, paragraph 3: 'infrared-save' -> safe p.5, paragraph 2, 'right of Fig. 2' -> Should be fig 3. p. 10, paragraph 1: ' BDT and LOLA indeed relying' -> BDT and LOLA are indeed relying

---

## Round 1 · Referee Report · Anonymous (Referee 3) · 2019-4-10

Strengths

1 - pragmatic approach that allows to apply machine learning based quark-gluon tagging to real phenomenological problems

Weaknesses

1 - The discussion of the observables in section 2.1 and of the 'LoLa' approach in section 2.2 is rather confusing and by no means self-contained.

2 - There are several confusing/misleading statements in the paper that should be clarified.

3 - Both phenomenological applications are rather superficial. Limitations should be highlighted.

4 - There are a few typos/referencing errors that need to be addressed.

Report

In the paper 'Quark-Gluon Tagging: Machine Learning meets Reality' the authors present an analysis of the notoriously difficult subject of quark-gluon tagging in the context of LHC phenomenology. The results of the paper are twofold: 1) an observable-based BDT is compared against a low-level input NN investigating the importance of detector effects in quark-gluon tagging, 2) the low-level input NN is applied to two phenomenological challenges at the LHC investigating possible improvements thanks to quark-gluon tagging. The content of this study is certainly topical and deserves publication. The phenomenological analysis in 2) unfortunately remains rather superficial and might only give an indication of possible improvements thanks to quark-gluon tagging. Still, after considering the minor corrections listed below, the paper should be published as it might serve as a motivation for further more detailed studies.

Requested changes

  • "new LoLa tagger" in the abstract should be clarified. The concept of a "LoLa tagger" is not commonly known to the interested reader. The same holds for "We will use our LoLa framework" in the Introduction.

  • In the Introduction the motivation for considering the monojet and dijet analyses appears rather arbitrary. This should be ameliorated.

  • the statement "We do, however, find that switching from Pythia to Sherpa makes quark–gluon discrimination generally a little harder." at the beginning of Section 2 should be clarified.

  • The discussion of the standard observables in Section 2.1 is rather confusing and should be streamlined.

  • The reference to Fig. 2 in the sentence "To determine what information the BDT is actually using, we plot the feature importance of each input variable on the right of Fig. 2. " in Section 2.1 should be for Fig. 3 I believe.

  • In Fig. 7 (right) the dependence on pminT,j is plotted. What is the definition of this observable? The caption only refers to pT,j which is not well defined for the multijet simulations considered here.

  • The simulation of H+jet and Z+jet in Section 3 is based on LO multi-jet merging in Sherpa. In Fig. 7 the "gluon fraction" of these samples is shown. It should be clarified how this "gluon fraction" is determined for a multijet final state.

  • In combination with the last point: it should be clarified how the truth information of a multi-jet final-state is obtained for the training of the LoLa tagger.

  • In Section 3 it is stated "A cut-and-count analysis above a stringent E/T requirement is not an optimal analysis strategy, because the small difference between the Higgs and Z masses hardly affects the kinematics." Nevertheless, as shown in Fig. 8 stringent cuts on MET improve S/B by an order of magnitude - and significantly more for light mediators as stated in the text. Thus, the effect of quark-gluon tagging of the recoling jet(s) should if possible also be investigated subject to a harsh MET cut.

  • The improvement of the dijet analysis thanks to quark-gluon tagging of both jets is very impressive. The authors suggest that quark-gluon tagging should already be applied at the trigger level. Is this realistic? What is the precise computational performance e.g. of the LoLa tagger?

  • The authors should discuss possible limitations of the presented Monojet and Dijet analyses.

---

## Round 1 · Referee Report · Anonymous (Referee 4) · 2019-4-14

Strengths

1- First application of the LoLa deep learning framework to q/g tagging. 2- Clear illustration of the importance of a pT-dependent training. 3- Motivates q/g tagging in two important LHC searches.

Weaknesses

1- Many aspects of the paper have been studied elsewhere with little or no mention in this paper. 2- There is no quantitative comparison of the LoLa method to other modern deep learning methods applied to q/g tagging. 3- It is not clear from the last section what the takeaway message is for q/g tagging in these two analyses.

Report

This paper reports a simulation study of q/g tagging and the application to two analyses at the LHC. q/g tagging is an important topic that has served as a benchmark for the jet tagging community and could have important implications for many analyses at the LHC. Therefore, this work is timely.

There are 4 parts to the paper: (1) the impact of detector-effects on q/g tagging, (2) the pT-dependence of q/g tagging, (3) the application of LoLa to q/g tagging, and (4) the application of q/g tagging to two analyses at the LHC. It is useful to consider each part because there is not a lot of overlap between parts and not all parts are novel.

For (1), this is not the first paper to use detector effects when studying deep learning and q/g tagging (see e.g. 1711.02633 for a pheno study and ATLAS/CMS notes for cases where full detector simulations are used; 1810.05165 also studies the implications of using detector 'safe' information). In particular, ATL-PHYS-PUB-2017-017 provides a direct comparison between detector-level and particle-level simulations. The current paper provides a more detailed comparison in Fig. 2b and Fig. 4 but I think it is important to discuss the connections with existing work.

It is mentioned many times that the observables used are not IRC safe. This is true, but is a bit jarring as it is repeated many times and not necessarily relevant. Just because an observable is not IRC safe does not mean that it cannot be powerful for q/g tagging (especially if one can train on data). Conversely, there has been a lot of interesting work to calculate IRC unsafe observable and to make new, similar observables that are IRC safe (see e.g. 1704.06266). I would suggest that you modify the language to reflect this. Related: there were qualitative statements made about comparisons between generators, but it would be helpful if this could be made concrete (i.e. given that IRC unsafe information is used, to what extent are the results consistent across generators).

For (2), it is well-known that the jet fragmentation depends on jet pT. The impact on deep learning has also been studied in the past in e.g. 1612.01551 and 1711.02633. However, I am not aware of doing a study that shows the impact of using a tagger trained at one jet pT and applied to another. This part of the paper is a nice study that shows the importance of using a jet-pT-dependent training. Could you add the rejection at a fixed efficiency in addition to the AUC? It is hard to gauge the impact from just the AUC numbers.

(3) is clearly completely new and the inclusion of charge is quite nice. However, it is unsatisfactory to not compare with other methods as has been done very thoroughly in e.g. 1810.05165 (compares DNN, CNN, RNN, and the new ones proposed in the paper). There is one sentence: "With our LoLa network we reproduce the performance of the enhanced images setup of Ref. [28] without detector simulation and after accounting for the move from Pythia to Sherpa." This is a lot to unpack and is quite qualitative. Please provide some quantitative statement about how this compares with other techniques (while it would be nice to do the complete comparison as in 1810.05165, even a partial comparison would be much better than nothing).

(4) After reading this section, I was not really sure what to conclude. For the dijet, I think one can conclude that a maximum ~20% improvement in the significance of a signal is possible with q/g tagging. Could you turn this into a statement about the current limits from ATLAS and CMS? I know you discuss the background uncertainty is unknown; it is okay to not estimate that but you can make a strong statement about the maximum improvement modulo background uncertainty. For the monojet, there is basically no quantitative conclusion and I'm not really sure what to take away.

Requested changes

1- Please change the title. I don't think it is representative of the work done in the paper. 2- Please provide the context for the detector-level studies you have performed (see the report for suggested references to add). 3- Please improve the discussion of IRC safety that is scattered throughout the paper (see report for details). 4- Add rejection at a fixed efficiency in addition to AUC numbers, as the latter are hard to interpret for the practical implication of q/g tagging. 5- Please make a quantitative comparison to other methods when showing the performance of LoLa. 6- For the dijet, please connect the 20% maximum improvement in sensitivity to the current CMS/ATLAS limits. 7- Please clarify the takeaway message for the monojet section (or remove it).

  • validity: good
  • significance: ok
  • originality: low
  • clarity: good
  • formatting: good
  • grammar: good

Author:  Jennifer Thompson  on 2019-05-21  [id 523]

(in reply to Report 4 on 2019-04-14)
Category:
remark
answer to question

We state in this paper that, with the same event generator and without detector simulation, we see the same performance as in our Ref. (31). Here we attach 2 of our preliminary plots, which compare our set-up with that of the 200-220 GeV jet sample in Ref. (31).

Attachment:

comparison.pdf

---

## Round 2 · Author Response

First of all, we would like to thank the referees for their careful
reading and for their comments. We agree that this paper essentially
has a negative bottom line, and we also agree that we cannot
enthusiastically report the impact of quark-gluon tagging on a
reference process. However, we believe that these finding should be
out in the public, and if only to encourage others to do better or to
drop quark-gluon tagging from the problems worth pursuing for LHC
searches.
reading and for their comments. We agree that this paper essentially
has a negative bottom line, and we also agree that we cannot
enthusiastically report the impact of quark-gluon tagging on a
reference process. However, we believe that these finding should be
out in the public, and if only to encourage others to do better or to
drop quark-gluon tagging from the problems worth pursuing for LHC
searches.

---

## Round 2 · List of Changes

Because many of the referees addressed the same points, we have collected them together for a combined response.
- We have changed the title to "Quark-Gluon Tagging: Machine Learning vs Detector".
- We added many references on quark-gluon tagging and on IR safety.
- We now refer to 1711.02633 for a similar study.
- We specified LoLa in the abstract and the introduction.
- We slightly modified the introduction of the two reference processes.
- We added a reference to the difference of q-g tagging based on Pythia vs Sherpa.
- We streamlined Sec.2.1.
- We expanded the discussion of Fig.2, relating our findings to the available literature.
- We expanded the discussion of Fig.3, including the relation to the incomplete and non-basis of observables.
- We added discussion of IR safety at the end of Sec.2.1 and now explicitly mention girth and C2 as safe.
- We now refer to the detailed comparison of architectures from 1810.05165 and make it clear that our new focus is on detector effects. We do not believe that a comparison to more than the CNN of 1612.01551 would add to the conclusions of the paper.
- We clarified the caption of Fig.7, how we extract MC truth from our simiulation, and the discussion of Fig.8.
- We unfortunately have no way of estimating the exact effect of quark-gluon tagging on a specific di-jet resonance search, but we relate it to event-level observables and a similar analysis.
- We have added a (blunt) bottom line to the mono-jet discussion, but we believe that the discussion should be kept in spite of the negative conclusion.
- We added a couple of references and clarifications as requested by the referees, including on pile-up.
A couple of points we could not change are:
- In the discussion of Fig.9 we already discuss the fact that for a stiff MET cut the quark-gluon tagging performance suffers.
- We are sorry, but adding reliable rejection efficiencies to Tab.1 would require us to use much more GPU power than we have. But an example number is 1/FPR=9.3 @ TPR=0.3.

---

## Editorial Decision

published